# When Bigger is Better: Revisiting Large-Batch Optimization in Language Model Pretraining

## Abstract

Large-batch training sizes promise near-linear speedups in language model pertain-
ing, yet existing studies highlight its poor optimization dynamics and degraded final
performance. In this paper, we seek to understand the failure of large-batch train-
ing, and show that it can in fact substantially outperform conventional small-batch
training. We first identify a critical oversight in the conventional view: large-batch
training can substantially surpass small-batch baselines when provided sufficient
tokens, but this advantage is often unrecognized due to its initial poor optimization
dynamics, manifested as larger gradient norms and even worse per-step loss during
early warm-up phases. To address this, we introduce a simple batch size scheduler
that stabilizes and improves training at remarkably large batch sizes. Our scheduler
scales pretraining up to batches of 32M tokens, using $3.3\times$ fewer computes to
achieve the superior later-stage performance of large-batch training. Detailed analy-
ses on gradient dynamics reveal that batch size fundamentally changes optimization
geometry. Notably, we show that classic gradient noise scale metrics fail to predict
the optimal batch size. Our findings offer practical recipes for designing efficient
and effective pretraining pipelines, and deepen the theoretical understanding of
large-batch optimization dynamics in language model pre-training.

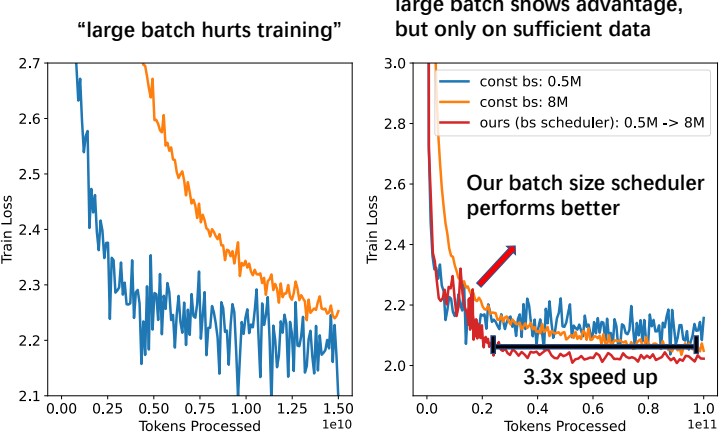

Figure 1: Training loss curve with different constant batch sizes and a batch size scheduler. **Left**: training loss curve up to 30B tokens; **Right**: training loss curve up to 100B tokens.

# 1  Introduction

Training language models with extremely large batch sizes unlocks near-linear speedups through data parallelism, slashing wall-clock communication time on multi-node clusters — an advantage that is especially pronounced for Mixture-of-Experts models [31, 5]. Yet, today's empirical evidence is discouraging: large-batch training frequently results in worse optimization dynamics and substantially degraded final model performance [15, 27, 20].

In this paper, we seek to understand the purported failure of large-batch training, and show that large-batch training is in fact viable. Specifically, we find that with a simple batch size scheduler, large-batch size training can substantially surpass the performance of small-batch baselines.

We first re-examine the traditional "large batch hurts training" results (Figure 1 left) and reveal its blind spot: when training on a sufficient amount of data, even naive constant large-batch size training shows clear advantages (Figure 1 right). Recent research on batch size scaling laws also supports this observation with the finding that optimal batch size scales primarily with data size [17, 32].

Taking a closer look at the training dynamics, we find that large batch sizes indeed impede optimization in the early training stage. It may be expected that large batch sizes will converge more slowly in terms of token efficiency due to fewer total optimization steps. However, counterintuitively, we find large-batch training is worse even in terms of optimization steps, despite consuming substantially more tokens than the small-batch baseline. Further analysis shows that large-batch training exhibits unstable gradient norms in the early stage.

To address this, we study a straightforward batch size scheduler: training with small batch sizes in the beginning to exploit its superior early optimization dynamics, then gradually increasing to the target large batch sizes to fully leverage its efficiency advantage. Specifically, we consistently improve training with a batch size of 8M tokens from the early stages through to completion. For extremely large batch sizes (e.g., 32M tokens), which initially fail within a given training budgets (e.g., 100B tokens), our batch size scheduler successfully enables effective large batch training.

Through detailed analyses of optimization metrics such as gradient norms, gradient noise, and optimizer update direction, we offer explanations for why different batch size schedules ultimately converge to similar final losses via a stabilization statement. We observe that classic gradient noise scale [20] metrics fail to predict optimal batch sizes accurately, highlighting the need for new metrics or insights into large-batch optimization dynamics.

Overall, our findings provide practical guidelines for designing more efficient pretraining strategies and deepen our theoretical understanding of how large-batch training dynamics influence language model pretraining.

# 2  Background and Experiment Setup

Batch size is an important hyperparameter in deep learning, yet its optimal tuning remains unclear [24, 8]. Some early studies argue that large batch training can hinder model performance, particularly in terms of generalization, suggesting that overly large batch sizes may be suboptimal. However, recent studies propose that the critical batch size [20]—the maximum batch size that maintains computational efficiency—scales with increasing data size. This implies that in contemporary large-scale pretraining scenarios, larger batch sizes might be preferable. Despite this, determining the optimal batch size remains an open question.

**Large batch training can hinder model generalization.**  Using excessively large batch sizes in training deep neural networks can negatively impact model generalization. Keskar et al.[15] observed that large-batch training tends to converge to sharp minima, which generally exhibit poorer generalization compared to the flatter minima associated with smaller batch sizes. Takase et al.[29] explained this by noting that reduced gradient noise in large-batch training restricts the model's ability to escape narrow minima. Similarly, Oyedotun et al. [22] argued that large batch sizes might lead to near-rank deficiencies in activation tensors, thereby adversely affecting the optimization process and generalization capability.

**Optimal batch size scales with the data size.**  On the other hand, recent research has started to explore batch scaling laws, examining how batch size relates to model size, data size, and compute.

Notably, Li et al.[17] and Zhang et al.[32] concurrently found that the optimal or critical batch size primarily depends on the amount of data rather than model size. Consequently, as training configurations scale up, larger batch sizes tend to offer better optimization.

**Comparing training loss curves between small and large batches.** Batch size comparisons typically focus on two metrics: per-token and per-step performance. The per-token axis measures loss against processed tokens, while the per-step axis measures loss against update steps. Commonly, large batch sizes perform better per-step due to more accurate gradient estimates but worse per-token due to fewer updates. However, the observed crossover in per-token performance highlights that early-stage results can be misleading. This finding prompts us to reconsider how to fairly evaluate small versus large batch sizes.

## 2.1 Setup

**Experiment Setup.** We train a series of auto-regressive causal language models in 164M. We set the number of Transformer layers to 12 and the hidden dimension to 768. We use a context length of 1024, SwiGLU MLP [25], Rotary positional embedding [28], RMSNorm, and untied embedding parameters. We train our model on the Pile dataset [6] with different token budgets, and we adopt the GPT-2 tokenizer. We use the AdamW optimizer [16, 19] with fixed hyperparameters $\beta_1$=0.9, $\beta_2$=0.95, a (coupled) weight decay of 0.1, and a gradient clipping of 1. We use batch size (BS) ranging from 0.5M to 32M. We use a warm-up and stable learning rate schedule by default. We use the fixed data amount strategy in warm-up phase for different BS by default. We use 1B tokens in warm-up for 100B budget, and 0.3B tokens for 30B budget. We train on 30B tokens in the grid search experiments and 100B for other experiments (majority). We use 4090, H200 GPUs for our experiments.

**Notation.** Let $\mathcal{D}$ be the data distribution and $L(\mathbf{w}, \mathbf{x})$ be the loss function where $\mathbf{w}$ denotes the model parameters and $\mathbf{x}$ denotes one sequence sampled in $\mathcal{D}$. Let $\tilde{\mathbf{g}}_{\mathbf{x}} = \nabla L(\mathbf{w}, \mathbf{x})$ be the (stochastic) gradient with one sequence in $\mathcal{D}$ [1]. We remove the dependence on $\mathbf{x}$ where it does not matter in the context. Let $\mathbf{g} := \mathbb{E}_{\mathbf{x}}[\nabla L(\mathbf{w}, \mathbf{x})]$ be the population gradient, and we have $\mathbf{g} := \mathbb{E}[\tilde{\mathbf{g}}]$. Then, the *population gradient norm square* is $\|\mathbf{g}\|^2$. Let the gradient noise covariance matrix be $\Sigma := \mathbb{E}_{\mathbf{x}}[(\tilde{\mathbf{g}} - \mathbf{g})(\tilde{\mathbf{g}} - \mathbf{g})^\top]$ and we mainly care about its trace $\mathrm{tr}(\Sigma)$, and call it *gradient noise*. The *gradient noise scale* [20] is defined as $\mathcal{B}_{\mathrm{simple}} := \mathrm{tr}(\Sigma)/\|\mathbf{g}\|^2$. Denote the first moment and second moment in Adam by $\mathbf{m}$ and $\mathbf{v}$, respectively. The Adam update direction is defined by $\mathbf{u} := \mathbf{m}/\sqrt{\mathbf{v} + \varepsilon}$, where the division is element-wise.

# 3 Analyzing the Failure of Large Batch Optimization

In this section, we seek to analyze and locate the failure of large-batch training. We first systematically verify that large batch size optimization has a significant performance degradation in the early stage (Sec. 3.1). Next, we focus on analyzing large-batch optimization dynamics in the warm-up phase, which is important for early-stage training (Sec. 3.2). Based on the experiments on different warm-up strategies, we identify that it is not suitable to do warm-up with a large batch size.

## 3.1 Large-batch Optimization Fails in the Early Stage

As illustrated in Figure 1, although large-batch training can eventually surpass the small-batch baseline once consuming sufficient training tokens, it lags far behind in the early stage before the two loss curves cross[2].

To confirm this is not due to our bad hyperparameter setup for the large-batch setting, we conduct a 2-D grid search over batch size and learning rate with 30B training tokens. We focus on tuning learning rates based on the hyperparameter scaling: the optimal learning rate often significantly changes with batch size [2]. Figure 2 left presents the results. Acorss the entire grid, every large-batch run is consistently and substantially worse than small-batch baselines.

---

[1]In most of the time, we consider the token-level batch size. But, when we estimate the gradient noise and gradient norm, we use the sequence-level batch size instead of token-level batch size There is basically a scale difference between sequence-level and token-level estimators.

[2]The 'early stage' here can be understood as the time before a critical point when the loss curve of large BS and small BS cross over. We will discuss more about the properties of 'early stage' in Section 4.4.

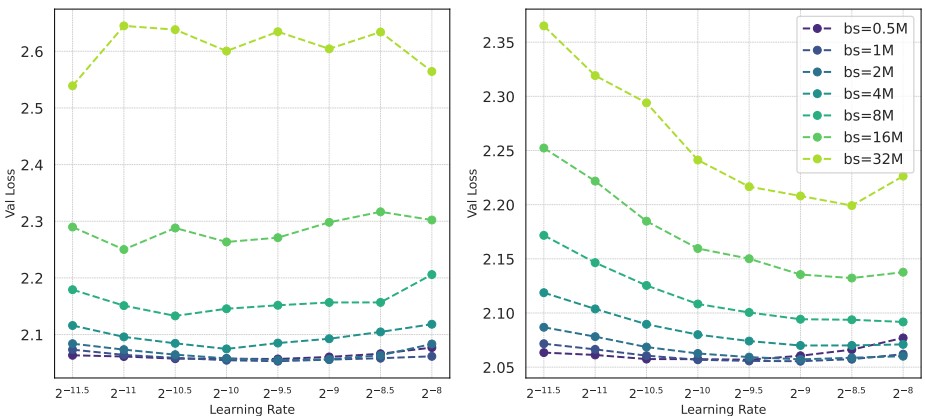

Figure 2: Validation loss w.r.t. batch size and learning rate with 30B tokens. **(a)**: constant batch size; **(b)**: using small batch (0.5M) in the warm-up and switch back to large batch size after warm-up.

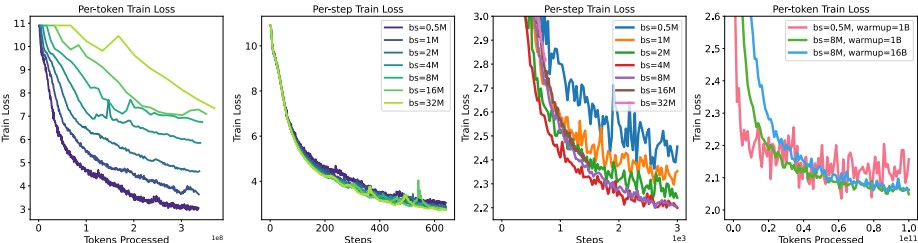

Figure 3: **(a)**: Training loss curve w.r.t. batch size with the same warm-up tokens. **(b)**: Training loss curve w.r.t. batch size with the same warm-up steps. **(a)(b)** share the same legend. **(c)**: Training loss curve w.r.t. batch size in per-step axis. A 0.3B fixed-token warm-up is used. **(d)**: Training loss curve comparing fixed-step and fixed-token warm-up. Comapred to 0.5M BS with 1B warm-up tokens, 8M BS + 1B warm-up is fixed-token strategy, 8M BS + 16B warm-up is fixed-step strategy.

A natural explanation is that, a fixed token budget gives far fewer updates to a large batch size: on 30B tokens, a batch size of 0.5M yields roughly 57,000 optimization steps, while a batch size of 32M yields only 900 steps. One may therefore expect a large batch size to shine once the optimization step is equalized due to its more accurate gradients. However, empirical results are in fact counterintuitive: as shown in Figure 3(c), even when we match the number of optimization steps, large-batch training still does not necessarily perform better than small-batch baselines. For example, a batch size of 4M tokens yields the best per-step loss curves, even outperforming the counterparts of larger batch — despite consuming up to $8\times$ more tokens.

Taken together, these results indicate a clear degradation of large-batch optimization in the early stage of training.

## 3.2 A Closer Look at the Warm-Up Phase

Warm-up is widely recognized as critical for stabilizing early optimization dynamics [7]. We therefore take a closer look at this phase to further analyze the failure of large-batch training.

**Fixed-token Warm-Up** We first experiment with 0.3B warm-up tokens across different batch sizes. Under this setting, the number of steps during warm-up scales inversely with the batch size. For example, while 0.3B warm-up tokens translate to 600 steps with a batch size of 0.5M, it is only about 10 steps with a batch size of 32M. Results are shown in Figure 3. As we can see, large-batch training exits the warmup phase with a significantly higher loss than small-batch baselines. Besides loss, the drawback of fixed-token warm-up for large-batch training also exhibits in gradients. As shown in Figure 4 (a), the gradient of large-batch training has not stabilized yet after warm-up phase, and is

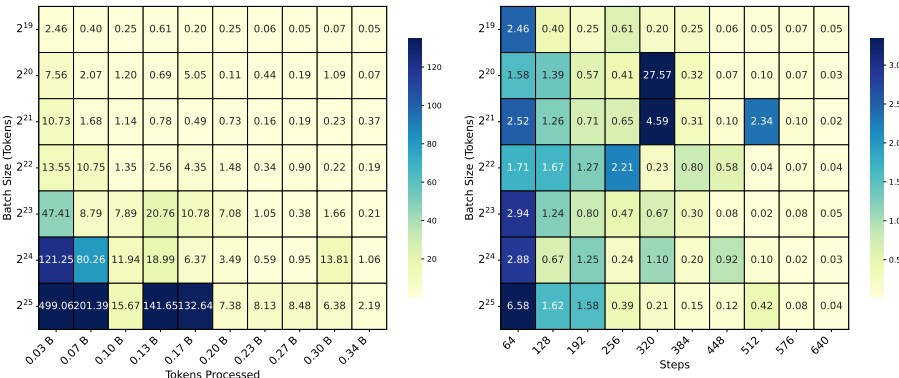

Figure 4: Population gradient norm square w.r.t. batch size with the same warm-up tokens (**Left**) and the same warm-up steps (**Right**).

still about $10\times$ bigger than the small-batch baseline. Thus, under a fixed-token setting, warm-up contributes far less to large batches than to small ones.

**Fixed-step Warm-Up** What if we instead equalize the steps in warm-up across different batch sizes? As shown in Figure 4 (b), doing so aligns the per-step loss curves across batch sizes. Moreover, as shown in Figure 4 (b), gradient norm all converges in a similar speed and to the same order or magnitude, and stabilizes.

However, this remedy is costly: the fixed step warm-up strategy will consume many more tokens for large batch sizes. For example, to match 600 warm-up steps, a batch size of 32M would consume 64B tokens. This might be more than the total token budget, or leave us many few tokens for the remaining primary optimization stage.

Furthermore, while fixed-step warm-up achieves better performance in the warm-up phase than fixed-token warm-up, we find that it leads to worse performance in the later training stage. Specifically, as shown in Figure 3 (d), when training with a large batch size of 8M tokens, the fixed-step warm-up setting (blue line) converges slower than the fixed-data warm-up setting (green line).

## 3.3 Summary

Based on our experiment results, we conclude that large-batch optimization fails at the early stage of training, and warm-up is one of the important factors that causes significant impacts on the performance of large-batch training.

However, both fixed-token and fixed-step warm-up schedules show inherent limitations. In short, it is ill-advised to warm up with a large batch size.

## 4 Batch Size Scheduler Unlocks Effective Large-Batch Training

Large batch sizes significantly improve large-scale training efficiency and boost final performance given sufficient data. Yet they often lag behind smaller batch sizes in the early training stage, and it is hard to mitigate their inherent optimization problem simply via tweaking early-stage training hyperparameters.

In this section, we show how a *batch size scheduler* delivers the best of both worlds: fast early progress reminiscent of small batches and the strong asymptotic performance of large batches.

We firstly start with a preliminary example that replace only the warm-up phase with small batch size and show its effectiveness. Then, we introduce a *linear batch size scheduler* can be competitive with both small batch training in the early stage and large batch training in terms of final performance. Moreover, batch size scheduler can effectively enable an extremely large batch training where the straightforward constant batch size degrades. Finally, we give a possible explanation in the perspective of hyperparameter and training process.

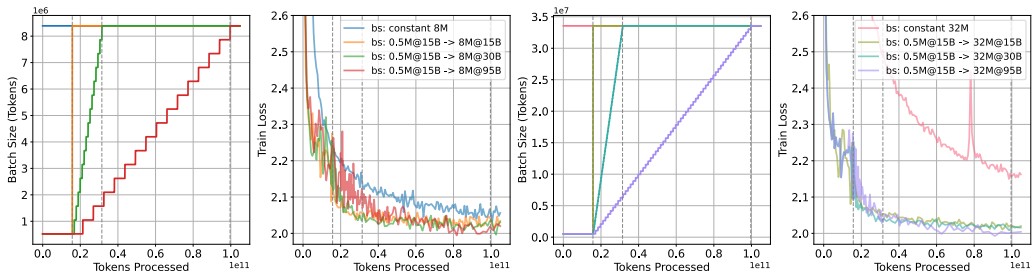

Figure 5: **(a)**: Batch size scheduler from 0.5M to 8M. **(b)**: Training loss curve with constant batch size and various batch size schedulers (8M). **(c)**: Batch size scheduler from 0.5M to 32M. **(d)**: Training loss curve with constant batch size and various batch size schedulers (32M).

### 4.1 Preliminary Study: Warm-Up with Small Batch Size

In Sec. 3, we find that large-batch training suffers in the initial warm-up phase, and merely tuning the warm-up length cannot fix this issue entirely. One straightforward idea is thus to do warm-up with a small batch size first, and then switch to the target large batch size.

To verify this idea, we conduct a 2-D grid search over batch size (BS) and learning rate (LR) with a 30B token budget. We replace the warm-up batch size with 0.5M while keeping a constant large batch size thereafter. Results are shown in Figure 2 (b). Compared to the constant large batch size baseline, starting a small batch size during warm-up consistently and substantially improves the final performance when the large batch size exceeds 8M across almost all learning rates. For example, for the largest batch size of 32M, by doing warm-up with the small batch size, we improve the best validation loss from 2.54 to 2.20.

### 4.2 Batch Size Scheduler

Replacing warm-up phase with small BS has achieved a significant improvement. In this section, we show that introducing a simple batch size scheduler strategy can further improves the large batch training. A simple linear batch size scheduler make large batch both competitive with small batch in the early and keep the advantage of large batch at convergence.

Similar to a learning rate scheduler, a batch size scheduler works by adjusting the batch size during the training process. In this work, we focus on linear batch size schedulers. Formally, given an initial batch size $B_{\text{init}}$, a target batch size $B_{\text{target}}$, a start token count $P$, and a ramp length $E$, we lineary interpolate the batch size from $B_{\text{init}}$ at $P$ tokens to $B_{\text{target}}$ at $P + E$ tokens. Figure 5 (a) and (c) shows all the batch size schedulers that we use in the experiments. As shown in Figure 5 (a), the scheduler divides $E$ tokens into equal-sized segments and uses a single batch size at each segment.

### 4.3 Results

Staring with an initial batch size of 0.5M, we begin to increase the batch size at 15B tokens; the final target batch size is 8M. We test three different ramp lengths $E \in \{0B, 15B, 80B\}$ (Figure 5 (a)).

**Finding 1: Linear batch size schedulers work consistently well.** Figure 5 (b) presents the results. We find all three linear batch size schedulers perform well in the early stage, addressing the failure in large-batch training. Then, they all converge to a similar final performance, slightly better than the constant BS baseline, keeping the advantage of large-batch optimization.

**Finding 2: Batch size schedulers enable extremely large batch training.** Furthermore, for a extremely large BS like 32M, unlike 8M, a straightforward constant BS training with 1B warm-up tokens does not perform well at the 100B token budget. However, the BS scheduler can enable the large batch training with 32M also enables the use of 32M, making extremely large batch training possible (Figure 5 (d)).

**Finding 3: The 0B ramp length works well without instability.** Surprisingly, we find a 0B horizon of increase, which means the BS is switched to 8M immediately from 0.5M, also performs well.

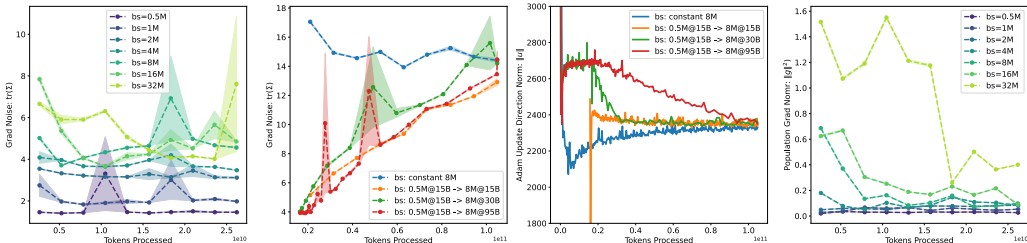

Figure 6: **(a)**: Gradient noise dynamics w.r.t. batch sizes (constant schedule) at the same learning rate. **(b)**: Gradient noise dynamics with baseline (8M) and various batch size schedulers. **(c)**: Adam update direction dynamics with baseline (8M) and various batch size schedulers. **(d)**: Population gradient norm dynamics w.r.t. batch sizes (constant schedule) at the same learning rate.

A suddenly drastic change of BS can make the gradient a big change, which could possibly cause instability problem in the training process. However, experiments show this aggressive strategy works well. Due to the ease of implementation, the 0B ramp length has more potential into practical use.

In Sec. 4.4, we explain that the target batch size determines local optimization geometry, such as gradient norms and noise levels, whereas the specific schedule shape has minimal impact. This explains why even a schedule with sudden jump (0B ramp length) can perform well, motivating our choice of a simple batch size scheduler.

## 4.4 A possible explanation

In this section, we offer a possible explanation for why different batch size schedules ultimately converge to similar final losses via a stabilization statement. Generally speaking, The stabilization statement argues that the local optimization geometry is determined by current hyperparameters and enough training time.

**The optimization geometry can adapt to the hyperparameters.** When we use fixed hyperparameters[3] and train the model for a certain number of tokens, the local optimization geometry will adapt to this hyperparameter stabilize. More specifically, we suspect there is a pre-stabilized stage and a stabilized stage during the model training. In the stabilization stage, the gradient-related quantities – including stochastic gradient noise, population gradient norm, first moment, second moment, and the update direction in Adam – will stabilize to a value determined by hyperparameters. And finally, the loss plateau in stabilization phase.

We provide evidence to our argument: In Figure 6 (a), we use the gradient noise $\mathrm{tr}\,(\Sigma)$ as an example to show that in each hyperparameter configuration, $\mathrm{tr}\,(\Sigma)$ does stabilize, and the stabilized value depends on hyperparameters (BS in this case).

**The stabilization is universal to training history.** We observe that this stabilization property does not depend significantly on the training history. Specifically, regardless of previous hyperparameter choices or scheduling strategies, the stabilized optimization geometry becomes consistent after training for enough tokens using the same final hyperparameters.

Empirically, we confirm this through two experiments comparing optimization metrics across batch size schedulers with the same final batch size. Figure 6 (b) shows the gradient noise $\mathrm{tr}\,(\Sigma)$. The constant batch size baseline maintains a stable gradient noise from 15B to 100B tokens. In contrast, schedulers initially exhibit increasing noise as the batch size ramps up, eventually stabilizing at the baseline level. Figure 6 (c) shows the norm of Adam update direction $\|\mathbf{u}\|$. Again, all schedulers eventually stabilize at the baseline value. However, the scheduler shape affects stabilization speed: a slower increase in batch size results in smoother stabilization.

**The length of pre-stabilization stage varies.** Now, we focus on the pre-stabilization stage. The length of this stage can vary significantly between different batch sizes. We observe that introducing a large batch size too early prolongs this phase.

---

[3]We mainly consider learning rate and batch size here since other hyperparameters like weight decay, momentum coefficient in Adam are fixed by default during the training.

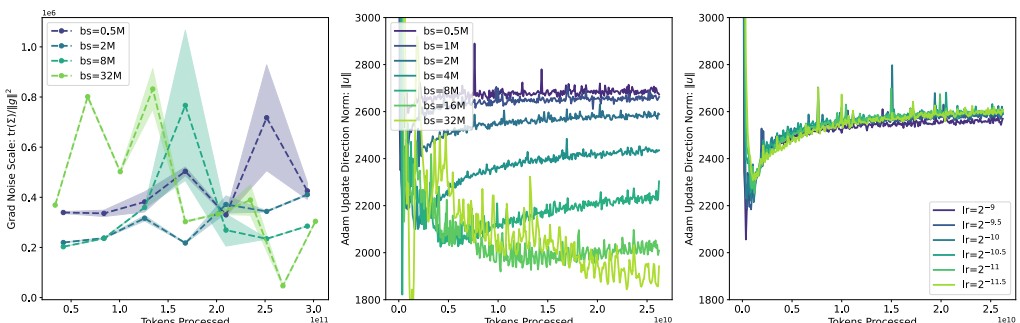

Figure 7: **(a)**: Gradient noise scale (the predictor for critical batch size) dynamics w.r.t. batch sizes at the same learning rate. **(b)**: Adam update direction dynamics w.r.t. batch sizes at the same learning rate. **(c)**: Adam update direction dynamics w.r.t. learning rates at the same batch size.

Figure 6 (d) shows that the gradient norm stabilizes faster for smaller batch sizes than for larger ones, especially at a batch size of 32M. This indicates the poor early-stage performance with large batches may result from slower stabilization, and using smaller batch sizes initially can accelerate stabilization.

**Summary.** In this section, we propose a stabilization perspective to explain training dynamics and optimization geometry under batch size scheduling. In the pre-stabilized stage, small batch size helps to stabilize faster, and make the early stage performance better. After transitioning to a large batch size and training further, the model adapts and benefits from the improved optimization efficiency of large-batch training. Additionally, our stabilization perspective implies that the exact speed of batch size scheduler is less important over a sufficiently long training period.

## 5 Discussion

In this section, we discuss more findings and insights from our experiments beyond the implications on batch size scheduler strategy.

**Gradient noise scale cannot predict critical batch size.** McCandlish et al. [20] introduces the gradient noise scale (GNS), defined in Section 4.4, and argue that it can be used to predict the critical batch size (CBS). They also mentioned that GNS depends on the learning rate via a "temperature" mechanism, and claimed that GNS prescribes an optimal batch size at any given temperature. To verify this point, we use a linear warm-up and constant learning rate (LR), use various batch size (BS) ranging from 0.5M to 32M, train on 300B tokens, and compute the GNS for intermediate checkpoints. In Figure 7 (a), we firstly see that GNS gradually stabilizes with some variance, which verifies our stabilization statement in Section 4.4 again. Surprisingly, we find the when using a BS greater than or equal to 2M, the stabilized value is far less than the BS used, indicating that predicted CBS is around 0.3M-0.5M and a BS larger than 2M should exhibit a significant performance degradation. However, the large BS will eventually surpass the small BS in performance as tokens processed increase even in the per-token axis (recall Figure 1). Additionally, the stabilized GNS decreases as the BS increase. This means the GNS cannot predict CBS in a straightforward manner and the relationship between GNS and CBS needs rethinking.

**The implicit bias of Adam: update direction primarily adapts to batch size.** As we show in Section 4.4, when we train the model with fixed LR and BS for at least a certain amount of data, the gradient-related metric will adapt to this LR and BS, and stabilize. We further explore the functional relationship of the stabilized value to LR and BS. In Figure 7 (b)(c), we find the stabilized norm of Adam update direction $\|\mathbf{u}\|$, i.e., magnitude of update before multiplying the LR, primarily depends on BS and is almost independent of LR. This is unexpected since the stabilized gradient, both in terms of signal ($\|\mathbf{g}\|^2$) and noise (tr $(\Sigma)$), does depend on LR at a fixed BS, while the gradient induced Adam update are not. We attribute this phenomenon to a new form BS-related *implicit bias of Adam*.

Understanding this phenomenon will probably help us figure out the algorithmic impact on optimal BS of different optimizers.

## 6 Related Work

**Batch size ramp-up.** Early literature propose to increase the batch size during the training process in order to reduce the number of parameter updates [4] or replace the learning rate decay [27], to improve training efficiency. In these works, batch size ramping up to 524288 images per batch [4] can achieve similar accuracies to small batch sizes. However, they operate in a relatively outdated experiment setting. The most significant difference in settings is that they do multi-epoch training while currently people use one-pass training particularly in the language models pretraining.

Recently, many technical reports on language model pretraining utilize the batch size ramp-up technique. Specifically, Llama 3 [9] increases batch size from a initial 4M tokens to 8M at 252M tokens, and then to 16M at 2.87T tokens. DeepSeek-V3 [3] gradually increases batch size from 12.6M to 63M in the training of the first 469B tokens, and keeps 63M afterwards. MiniMax-01 [21] fits a batch size scaling law w.r.t. loss and doubles the batch size whenever loss reaches this fitted line. In the specific pre-training process, they increases batch size from a initial 16M to 32M at 69B tokens, then to 64M at 790B tokens, and finally to 128M at 4.7T tokens. MiniCPM [13] doubles the batch size from 2M to 4M at about 500B tokens. Compared to the original batch ramp-up, these works tend to do batch size ramp-up mainly in the *early stage* of training, and up to a *mild* global batch size. However, They do not offer a principled guideline about using this technique.

**Hyperparameter scaling laws.** Beyond the scaling laws of loss w.r.t. model size and data amount [14], and compute-optimal scaling laws [12], researchers start to study hyperparameter scaling laws [1, 2, 13, 17, 23, 26, 30]: the relationship between optimal hyperparameters – typically learning rate and batch size – and interested independent metric, including model size, data amount, compute, or even loss value. Serval works study the batch size scaling with expected loss value [14, 13, 30, 26], thus they cannot predict the optimal batch size a priori. Other work study optimal batch size as a function of model, data, or compute [1, 2, 17, 23]. Their results are typically like optimal learning rate becomes smaller and optimal batch size becomes larger when the compute budget increases.

**Critical batch size.** It was argued that there exists a critical batch size (CBS): increasing the batch size up to the CBS results in minimal degradation, while further increasing it beyond the CBS yields unneglectable performance degradation. Previous studies suggest that when the batch size is lower than this critical value, the learning rate needs to be proportionally adjusted according to batch size [8, 11]. McCandlish et al. [20] introduces the gradient noise scale, and argue that it can be used to predict the CBS. Follow-up works talk about the efficient computation for gradient noise scale [10] and the extension to Adam [18].

## 7 Limitations and Conclusion

**Limitations.** One limitation of our work is that we do not verify all of our findings on larger-scale language models such as 1B or 8B, due to our limited computational resources. However, we observe similar phenomenon based on our preliminary experiment results on 1B models, such as 1) poor early-stage optimization of large-batch training and 2) effectiveness of batch size scheduler. In addition, as shown in prior work about critical batch size, the optimal batch size is often independent of model parameters. We leave more analysis about our findings on larger models to future work. Another limitation of work is that we only focus on linear batch size schedule. Future work can study more sophisticated batch size scheduler following prior work on learning rate scheduler.

**Conclusion.** Enabling extremely large batch size pre-training is a fundamental problem for efficient modern language model pre-training. Existing empirical evidence highlights the poor optimization dynamics and the degraded final performance of large-batch training. However, through detailed analysis of the optimization dynamics of large-batch training over a long horizon, we show that 1) large-batch training mainly suffers from poor its early-stage optimization, but has superior performance in the later stage. The empirical success of our batch size scheduler, alongside our theoretical understanding of large-batch optimization dynamics, suggests that it is promising to enable more efficient and effective language model pre-training at extremely large batch size scales.

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
