# OpenReview forum: "When Bigger is Better: Revisiting Large-Batch Optimization in Language Model Pretraining"
_NeurIPS.cc/2025/Conference — Submitted to NeurIPS 2025_

### Official Review · Reviewer_4vk7 · 2025-06-22

**Clarity:** 3
**Significance:** 3
**Originality:** 3
**Rating:** 4
**Confidence:** 4

**Summary:**

This paper offers a new perspective on the observation that, contrary to prior belief, larger batch sizes can lead to better performance in language model training. The authors argue that the commonly observed underperformance of large batches is primarily due to poor behavior during the warmup phase, especially under limited data regimes. To address this, the paper proposes a novel warmup strategy that improves overall performance.

**Questions:**

1.	Do the conclusions hold for larger models? As model size increases, how does the behavior during the warmup phase change?
	2.	Can the proposed hypothesis generalize to tasks beyond NLP, such as computer vision? According to the paper, warmup plays a critical role, but prior work suggests that warmup may have limited impact in some domains. Does the proposed explanation still hold in those cases?
	3.	The idea of addressing training stability through batch size and learning rate schedules has been explored in works like “Don’t Decay the Learning Rate, Increase the Batch Size.” Why do those previous methods fail to produce similar benefits, and how is the proposed approach fundamentally different?

Given the hypothesis presented in this paper is thought-provoking, I am currently leaning toward a borderline accept. However, if other reviewers raise additional concerns, I am open to revising my score.

**Ethical Concerns:**

["NO or VERY MINOR ethics concerns only"]

**Final Justification:**

Thank you for your response. The clarification and experiments you provided have addressed some of my concerns. I will keep my score.

I noticed that other reviewers did not give particularly high scores. While I believe the problem explored in the paper is valuable and inspiring, the scale of the experiments—especially given the current size of large language models—is simply too small, even for a 1B model. Therefore, I do not intend to persuade the other reviewers, but I will maintain my score.

**Limitations:**

See weakness and questions.

**Paper Formatting Concerns:**

N.A.

**Quality:**

3

**Strengths And Weaknesses:**

The paper is well-written, with a clear flow from motivation to analysis, validation, and solution. The proposed hypothesis offers a compelling explanation for why large batch sizes can yield better performance in LLM training—contrary to earlier observations in other domains.

However, given the ongoing debate around this topic, the evidence provided so far is somewhat limited:
	1.	The experiments are conducted only on models with ~100M parameters, which is relatively small. Validation on larger models (at least >1B parameters) is necessary to better support the claims. It would also help to clarify how model scale interacts with the proposed warmup behavior.
	2.	The hypothesis should be tested across a broader range of tasks. In particular, prior observations of large batch degradation often come from computer vision. Testing the proposed explanation and warmup strategy in such CV tasks could help determine whether the findings generalize beyond NLP or Transformer-based architectures.

---

> ### Author Rebuttal · Authors · 2025-07-31
>
> Thank you for your careful review. Regarding your questions:
>
> > Q1. Experiments on larger (1B) model
>
> We have conducted our primary experiments (Figure 1) using a 1B model trained on 30B tokens, and the results are summarized in Table 1. To clearly illustrate the training loss curve in tabular form, we smooth the training loss and uniformly sample 17 points.
> Due to computational constraints, we employed a fixed learning rate (LR) of 2^-10, with a warmup phase of 1B tokens. Additionally, we linearly decayed the LR to 0.05x during the final 3B tokens. The overall trend for the 1B model closely matches Figure 1. **Our main conclusions remain valid for the 1B model**:
> - After the warmup phase, a significant performance gap appears between small BS (1M) and large BS (4M, 32M), indicating that large BS performs poorly during warmup and early training.
> - With sufficient training (around 26B tokens), large BS (4M) surpasses small BS (1M).
> - The BS scheduler outperforms small BS (1M) and successfully enables training with an extremely large BS (32M).
> - Regarding LR decay: although small BS (1M) benefits more from LR decay (notably reducing loss from 27B to 30B), our claim that **large BS exhibits an advantage given sufficient data holds true even with LR decay**. Please refer to the *Table 1 in our reply to Reviewer tybB* for additional details.
>
> Table 1. Smoothed training loss curve with 1B model.
>
> | Tokens (B) | BS=1M | BS=4M | BS=32M | BS scheduler (1M -> 32M)|
> |-|-|-|-|-|
> | 1 (warmup phase ends)| 2.962 | 4.839 | 7.470  | 2.940|
> | 2 | 2.349 | 3.146 | 6.797  | 2.340|
> | 4 | 2.105 | 2.370 | 5.912  | 2.099|
> | 6 | 2.027 | 2.158 | 5.261  | 2.021|
> | 8 | 1.984 | 2.068 | 4.802  | 1.978|
> | 10                   | 1.956 | 2.015 | 4.481  | 1.949                   |
> | 12 (BS growth begins)| 1.934 | 1.979 | 4.174  | 1.926                   |
> | 14                   | 1.919 | 1.952 | 3.896  | 1.874                   |
> | 16                   | 1.906 | 1.930 | 3.612  | 1.836                   |
> | 18                   | 1.894 | 1.912 | 3.394  | 1.819                   |
> | 20                   | 1.886 | 1.897 | 3.195  | 1.809                   |
> | 22| 1.878 | 1.884 | 3.053  | 1.802|
> | 24 (BS growth ends)  | 1.873 | 1.874 | 2.916  | 1.796|
> | 26 (large BS surpass small BS) | 1.865 | 1.862 | 2.810  | 1.789|
> | 27 (LR decay begins) | 1.865 | 1.860 | 2.762  | 1.790|
> | 28                   | 1.847 | 1.847 | 2.688  | 1.785|
> | 30                   | 1.787 | 1.806 | 2.585  | 1.777|
>
>
> > Q2. Experiments on CV tasks
>
> Due to time and computational constraints, we do not have additional results on CV tasks available during this rebuttal. However, we would like to add some remarks.
>
> To verify our observations in CV tasks, it is best to align all experimental settings except for the data (images). Specifically, training a transformer with the AdamW optimizer using a one-pass dataset is ideal. In this context, large-scale CLIP pretraining on image-text pairs would be particularly suitable. Unfortunately, downloading such a large dataset and completing experiments within one week was not feasible.
>
> Testing our observation under a more classical setting is also valuable. For example, we can consider training a ViT model on ImageNet for image classification, although in a multi-epoch scenario.
>
> We will share results during the discussion period as soon as they are available.
>
> > Q3. According to the paper, warmup plays a critical role, but prior work suggests that warmup may have limited impact in some domains. Does the proposed explanation still hold in those cases?
>
> Firstly, we'd like to clarify what "critical" means in our context and then discuss the literature that claims "warmup may have limited impact in some domains," and discuss the relationship with our findings.
>
> In our setting, the warmup stage is critical in the sense that large BS performs poorly during warmup and early training compared to small BS. However, when performing warmup with a small BS before switching immediately to a large BS, the performance of the large BS significantly improves, especially at early training.
>
> We find the limited impact of warmup in prior works can be intepreted in two ways:
>
> - **Perspective 1: Warmup may have limited impact with CNN + SGD compared to Transformer + Adam.** Early studies [6] indicate that warmup improves CNN + SGD performance, but training does not collapse without it. Meanwhile, [7] highlights that warmup stabilizes adaptive LR at early stages, and omitting it harms Transformer training.
>
> Currently, we do not have results for CNN + SGD to confirm whether large BS underperforms early with or without warmup or whether small BS warmup improves subsequent large BS training. We will provide these results when available.
>
> - **Perspective 2: Final performance is insensitive to exact warmup length.** Previous work [8] finds warmup length has limited impact on final performance (Figure 4 in [8]). Our experiments (*Table 3, reply to Reviewer tybB*) similarly show diminishing warmup impact as training proceeds. However, the consistently poor early-stage performance of large BS compared to small BS remains across varying warmup lengths, which suggest that our observation does not contradict the viewpoints presented in prior works.
>
> > Q4. The idea of addressing training stability through batch size and learning rate schedules has been explored in works like “Don’t Decay the Learning Rate, Increase the Batch Size.” Why do those previous methods fail to produce similar benefits, and how is the proposed approach fundamentally different?
>
> Our overall answer: technically, our proposed BS scheduler is not fundamentally different from previous methods [1,2]. However, the experimental settings and motivations differ significantly.
>
> - **What [1,2] exactly did:**
>     - Smith et al. [1] replace LR decay with increased BS during multi-epoch LR schedules, focusing on matching loss curves, not enhancing large BS training performance.
>     Specifically, if the original LR decay is set to 0.2x at some time, we would instead increase the BS by 5x and keep LR constant. They find the training loss curves can basically match.
>     - Devarakonda et al. [2] partially replace LR decay with BS increases to reduce parameter updates.
>     Also, they only increase BS when there is a LR drop and adapt LR decay accordingly.
>     Specifically, if the original LR decay is set to 0.1x, and we choose to increase the BS by 5x at this point, we would adjust the decay factor to 0.5x to compensate for this increase.
>     They find loss curves can match and constant large BS underperforms.
> - **Differences in setup (data, model, optimizer):**
> Previous works [1,2] focus on multi-epoch CNN image classification with SGD. Our work uses one-pass data, Transformers, and Adam optimizer. Earlier study [3] showing poor generalization with large BS is similar to settings in [1,2], while recent works [4,5] discussing optimal BS scaling are similar to our setting. \
> **Among many differences in setup, we suspect the one-pass or multi-epoch setting is a key factor to the conclusion.**
> To verify it, we conduct experiments on a 3B subset of pile and train in multiple epochs. Due to time constraints, we use the same settings as in Figure 1 (fixed LR of 2^-10, BS 0.5M and 8M) and compare them.
> We present results in Table 2.
> We can see:
>     - Large BS initially underperforms but eventually surpasses small BS in both one-pass and multi-epoch settings.
>     - The performance gap shrinks under multi-epoch conditions, which suggests dataset reuse might influence benefits of large BS.
>
>     Based on these experiments, we do observe the multi-epoch training can make some differences: the benefit of large BS become less pronounced in multi-epoch setting. However, since the optimal HPs in the multi-epoch setting may differ from the one-pass setting, we cannot currently make a definitive claim regarding the impact of BS.
>
> Table 2. Validation loss for various BS and data setup.
>
> | (BS, Data setup) \ Tokens (B) |10|20|30|40|50|60|70|
> |-|-| ----- | ----- | ----- | ----- | ----- | ----- |
> |(0.5M, multi-epoch)| 2.231 | 2.200 | 2.181 | 2.172 | 2.163 | 2.160 | 2.156 |
> |(8M, multi-epoch)| 2.353 | 2.241 | 2.193 | 2.171 | 2.154 | 2.142 | 2.142 |
> |(0.5M, one-pass) (in Figure 1)| 2.208 | 2.168 | 2.155 | 2.142 | 2.133 | 2.127 | 2.127 |
> |(8M, one-pass) (in Figure 1)| 2.321 | 2.196 | 2.141 | 2.115 | 2.097 | 2.084 | 2.073 |
>
>
> - **Difference in the use of BS scheduler and motivations.**
>     - Previous methods [1,2] focus on LR-BS interactions, increasing BS only when LR drops. They find a linear LR-BS scaling rule holds for SGD on CNN-based image classification tasks, matching loss curves up to a 0.1M batch size.
>     - However, we observe that while constant large BS initially performs poorly, it eventually outperforms small BS with sufficient training at a fixed LR. To address initial issues with large BS, we introduce a BS scheduler transitioning from small to large BS.
>     - In summary, prior work finds constant large BS ineffective and uses an LR-aware BS scheduler for improvement. We find constant large BS effective given sufficient training and use an LR-independent BS scheduler to bridge the early-stage gap.
>
> [1] Don’t Decay the Learning Rate, Increase the Batch Size, ICLR 2018.
>
> [2] AdaBatch: Adaptive Batch Sizes for Training Deep Neural Networks, 2017.
>
> [3] On Large-Batch Training for Deep Learning: Generalization Gap and Sharp Minima, ICLR 2017.
>
> [4] DeepSeek LLM: Scaling Open-Source Language Models with Longtermism, 2024.
>
> [5] Predictable Scale: Part I — Optimal Hyperparameter Scaling Law in Large Language Model Pretraining, 2025.
>
> [6] Accurate, Large Minibatch SGD: Training ImageNet in 1 Hou, 2017.
>
> [7] On the Variance of the Adaptive Learning Rate and Beyond, 2019.
>
> [8] Why Warmup the Learning Rate? Underlying Mechanisms and Improvement, 2024

---

> > ### Comment · Reviewer_4vk7 · 2025-08-01
> >
> > Thank you for your response. The clarification and experiments you provided have addressed some of my concerns. I will keep my score.
> >
> > I noticed that other reviewers did not give particularly high scores. While I believe the problem explored in the paper is valuable and inspiring, the scale of the experiments—especially given the current size of large language models—is simply too small, even for a 1B model. Therefore, I do not intend to persuade the other reviewers, but I will maintain my score.

---

### Official Review · Reviewer_tybB · 2025-06-30

**Clarity:** 2
**Significance:** 2
**Originality:** 2
**Rating:** 2
**Confidence:** 4

**Summary:**

This paper argues that batch ramp is crucial for obtaining competitive performance while minimizing the sequential number of steps for a 164m model. It conducts multiple experiments with sweeps across learning rate and batch size and argues that during warmup phase, a small batch size is crucial for stabilization.

**Questions:**

My main question to the authors is about using the warmup and stable schedule. Why was cosine decay, or some other form of decay not considered in the experiments? Also, in section 3.2, when discussing about warmup steps, was the learning rate tuned for different warmup durations? It is not clear if proper learning rate tuning was done for varying warmup strategies.

**Ethical Concerns:**

["NO or VERY MINOR ethics concerns only"]

**Final Justification:**

The authors did try to run some preliminary experiments with decay and beta_2 sweep. But, the paper is not yet ready for a publication and needs to seriously consider learning rate decay and hyper parameter tuning. Thus I am planning to keep my score.

**Limitations:**

I think the main limitation is the empirical setup in the work, which does not seem suitable for studying the question of batch size ramp up as stated before.

**Quality:**

1

**Strengths And Weaknesses:**

The premise of studying if batch ramp up is crucial is important. I also like the results about the stability of certain parameters related to optimization geometry, independent of the history. However, I think the empirical setup in the paper is not suitable enough for studying the batch size ramp up question. First of all, the learning rate schedule given by warmup and stable does not include decay and contributes to the worse performance of small batch sizes in long runs due to iterate variance. Moreover, various parameters like $\beta_1$ and $\beta_2$ of Adam are not tuned with batch size, which is known to be crucial for good performance at small batch sizes [1,2]. Based on these, I am concerned about the validity of most of the empirical results in the work.

[1] - Resolving Discrepancies in Compute-Optimal Scaling of Language Models 2024

[2] - How Does Critical Batch Size Scale in Pre-training? 2024

---

> ### Author Rebuttal · Authors · 2025-07-31
>
> Thank you for your constructive review. Regarding your questions:
>
> > Q1. Regarding learning rate (LR) schedule.
>
> Thank you for your question! Indeed, we did not explicitly explain why we used a constant LR in the main text, and we are happy to clarify it here.
>
> Firstly, we would like to emphasize that studying a constant LR is also meaningful.
> - **Compared with schedulers involving decay (including cosine and linear schedules), a constant LR has its unique advantages.** For a scheduler with decay, the points along the training trajectory are generally not equivalent [3]. Specifically, in a cosine scheduler designed for training 50B tokens, the 40B checkpoint obtained during training cannot directly represent the performance of training with only 40B tokens; instead, one should rerun the experiment with a separate cosine scheduler designed specifically for 40B tokens. \
> However, this issue does not exist with a constant LR scheduler, enabling us to perform more experiments with limited resources. Computational efficiency was thus the primary reason behind our choice of a constant learning rate schedule.
> - **Understanding constant LR clearly is a prerequisite for further research into other schedulers.** On one hand, it removes the additional complexity introduced by LR scheduling, allowing us to investigate the behavior in a minimal and straightforward setting. On the other hand, understanding the behavior of constant LR schedules helps us better understand more complicated schedulers. Indeed, many works studying learning rate schedulers start from constant LR experiments [4,5,6].
> - About the references you provided:
> Both [1] and [2] discuss batch size scaling laws.
> Porian et al. [1] found that by optimizer tuning, a constant LR could achieve performance comparable to a cosine scheduler. However, we acknowledge that they tuned more hyperparameters, including batch size (BS), LR, and \beta_2, whereas we used a fixed \beta_2 in our main experiments.
> Zhang et al. [2] studied the critical batch size under a constant LR + EMA setting.
> Both works at least suggest that a constant LR setup is not unsuitable to study.
>
> On the other hand, we completely agree that studying the effect of LR decay on BS is important. Therefore, we also chose the WSD scheduler, which closely relates to constant scheduler, to perform supplementary experiments.
> Specifically, for a constant LR run of 100B tokens, we applied LR decay starting from checkpoints at 30B, 40B, ..., 100B tokens. Due to time and computational constraints, we used a fixed peak LR of 2^−10 and a fixed LR decay strategy: linearly decaying to 0.1x peak LR over 5B tokens. Additionally, we provide results only with constant batch size. We present these results in Table 1 and mark the optimal BS for each training horizon.
>
> We can see:
> - Small BS benefit significantly from LR decay, and the advantage of larger BS becomes less pronounced compared to constant LR. We also provide a similar table for constant LR (*Table 3 in our reply to Reviewer haAx*).
> - However, we would like to emphasize that the trend where larger BS gradually become better than smaller BS still exists. We present two pieces of evidence for this:
>     - The optimal BS increases as we train with more tokens. Specifically, the optimal BS is 1M from 35B to 65B tokens and 2M from 75B to 105B tokens, although the rate of increase is not rapid.
>     - Larger BS gradually outperform smaller ones. Specifically, the 4M batch size outperforms 0.5M at 55B tokens, and the 8M batch size achieves almost the same loss as 0.5M at 105B tokens.
>
> Table 1. Validation loss for various BS and tokens with linear LR decay
> |BS \ Tokens (B)|35|45|55|65|75|85|95|105|
> |-|-|-|-|-|-|-|-|-|
> |0.5M|2.060|2.051|2.044|2.039|2.035|2.032|2.029|2.027|
> |1M|**2.050**|**2.039**|**2.032**|**2.025**|2.021|N/A|2.013|2.011|
> |2M|2.054|2.042|2.033|2.026|**2.020**|**2.016**|**2.012**|**2.009**|
> |4M|2.067|2.051|2.041|2.033|2.026|N/A|2.017|2.013|
> |8M|2.099|2.078|2.064|2.053|2.045|N/A|2.033|2.028|
> |16M|2.167|2.129|2.106|2.089|2.076|N/A|N/A|2.051|
> |32M|2.445|2.336|2.270|2.237|2.201|N/A|2.156|2.141|
>
> > Q2. Moreover, various parameters like and of Adam are not tuned with batch size, which is known to be crucial for good performance at small batch sizes [1,2].
>
> This is a good point. Many previous works have mentioned that the optimal \beta_2 of Adam scales with the BS. We first briefly review this statement:
>
> **The choice of $\beta_2$.**
> The references you provided [1,2] sweep \beta_2​ over selected values. Porian et al. [1] sweep over the grid [0.95, 0.99, 0.999], while Zhang et al. [2] use a larger grid [0.95, 0.99, 0.995, 0.999, 0.9995].
> Other works propose scaling rules for \beta_2 with respect to BS. Malladi et al. [7] propose scaling as $\beta'_2 = 1 - B'/B(1-\beta_2)$, and recently Marek et al. [8] propose scaling as $\beta'_2 = \beta_2^{(B'/B)}$. These two scaling rules coincide when we do a first-order Taylor expansion of the second rule around \beta_2=1. Since \beta_2​ is close to 1 in practice, we can approximately view these rules as equivalent. When the BS approaches 1, these rules suggest we should use a \beta_2 value very close to 1.
>
> **The improvment from tuning $\beta_2$.**
> Note that [1,2,8] all work in the Chinchilla setting, i.e., train each model of size N for 20N tokens. All these works find that a larger \beta_2 is better for smaller BS. We summarize their evidences below:
> - Zhang et al. [2] reports the optimal \beta_2 in their Table 4, but they do not specify how much improvement it brings. In Figure 8(b), they show a larger \beta_2 improves about 0.01-0.02 in validation loss with about 30B tokens (over 60k steps and with a BS of 0.5M(=1024x512)). We agree that this is a strong evidence to support the statement.
> - Porian et al. [1] report similar findings. Based on their Figure 15, we conclude that the advantage of a larger beta_2​ is more significant when BS is at or below 0.25M (=128x2048), up to a model size of 221M parameters. However, the training horizon in Figure 15 of [1] for the largest model (221M) is relatively short (only 4B tokens).
> - Marek et al. [8] provide evidence in their Figures 5 and 6. In Figure 5, they train a 19M model with about 400M tokens. The gain from using a larger \beta_2​ is very significant (around 0.2 loss) at a BS of 8K (16x512). In Figure 6, they show that for a 1.3B model trained for 10B tokens, increasing \beta_2​ from 0.95 to 0.9999 significantly improves the validation loss (about 0.8) at BS=1 (2K tokens). We also consider this strong evidence.
>
> Based on the literature, we strongly agree that tuning \beta_2​ is important for small BS. Additionally, we believe how much tuning \beta_2 can improve over a longer training horizon remains an open problem. Compared to the listed works, where tokens per parameter (TPP) is 20, we use significantly higher TPPs: 625 (100B tokens and 160M model) in Figure 1 and 187 (30B tokens and 160M model) in our sweeping experiments (Figure 2), which is in an over-trained setting.
>
> In summary, studying the effect of \beta_2 on BS in an over-trained setting is important yet remains an open problem. It requires significantly more computational resources to conduct thorough tuning experiments, which unfortunately cannot be completed exhaustively within this rebuttal period.
>
> Finally, we provide some preliminary experiments regarding the effect of \beta_2. We use the same setting as Figure 1 for the 0.5M BS, except that we use a larger \beta_2 of 0.995 and train for 60B tokens with a constant LR of 2^-10. We select this value based on the \beta_2 scaling rule [7,8], by substituting the original \beta_2=0.95 and B'/B=1/8 or 1/16. The results are presented in Table 2. We observe that:
>
> - A larger \beta_2 improves performance throughout the training. The improvement is more pronounced with longer training.
> - However, a large BS can still outperform a small BS with a large \beta_2, given sufficient training.
>
> Based on this experiment, we believe our main observation still holds even when tuning the \beta_2 for a small BS.
>
> Table 2. Validation loss for various BS and beta_2 in Adam.
> |(BS, beta_2) \ Tokens (B)|10|20|30|40|50|60|
> |-|-|-|-|-|-|-|
> |(0.5M,0.95)|2.208|2.168|2.155|2.142|2.133|2.127|
> |(0.5M,0.995)|2.202|2.156|2.141|2.123|2.121|2.113|
> |(8M,0.95)|2.321|2.196|2.141|2.115|2.097|2.084|
>
> > Q3. It is not clear if proper learning rate tuning was done for varying warmup strategies.
>
> Due to time and computational constraints, we have not jointly tuned LR and warmup. We would like to remark that the complexity of HP tuning grows exponentially with the number of HPs. Nonetheless, we provide experiments demonstrating that the warmup length used for the large batch size (8M) in our main experiments (Figure 1) is not poorly chosen. The results are presented in Table 3. This gives further evidence for our observation that large BS does not work well in warmup stage.
>
> Table 3. Validation loss for various LR and warmup length at BS=8M.
> |(LR, warmup) \ Tokens (B)|8B|16B|24B|
> |-|-|-|-|
> |(2^-9,0.3B)|2.513|2.304|2.231|
> |(2^-9,1B)|**2.399**|**2.247**|**2.184**|
> |(2^-9,16B)|2.579|2.312|2.195|
> |(2^-10,0.3B)|2.521|2.276|2.198|
> |(2^-10,1B)|**2.392**|**2.228**|**2.166**|
> |(2^-10,16B)|2.57 (10B)|2.27 (20B)|2.18 (30B)|
>
> [1] Resolving Discrepancies in Compute-Optimal Scaling of Language Models, NeurIPS 2024.
>
> [2] How Does Critical Batch Size Scale in Pre-training? ICLR 2025.
>
> [3] Training Compute-Optimal Large Language Models, NeurIPS 2022.
>
> [4] Scaling Law with Learning Rate Annealing, 2024.
>
> [5] A Multi-Power Law for Loss Curve Prediction Across Learning Rate Schedules, ICLR 2025
>
> [6] Scaling Collapse Reveals Universal Dynamics in Compute-Optimally Trained Neural Networks, 2025.
>
> [7] On the SDEs and Scaling Rules for Adaptive Gradient Algorithms, NeurIPS 2022.
>
> [8] Small Batch Size Training for Language Models: When Vanilla SGD Works, and Why Gradient Accumulation Is Wasteful, 2025.

---

> ### Comment · Reviewer_tybB · 2025-08-08
>
> Sorry for the delayed reply. Although the experiments provided in the rebuttal phase follow more standard training practices, but understanding batch size warmup requires properly tuning beta1, beta2 and learning rate decay, which has not been done for majority of the experiments. I think the authors need to revise the paper with proper hyper parameter tuning and the experiments provided in rebuttal are not sufficient. I therefore maintain my rating.

---

### Official Review · Reviewer_Q7C1 · 2025-06-30

**Clarity:** 2
**Significance:** 2
**Originality:** 1
**Rating:** 2
**Confidence:** 4

**Summary:**

This work revisits the large batch training problem in LLMs using gradient noise scale. A simple batch size scheduler to stabilize training is proposed.

**Questions:**

1.There’s loss spike shown in the Fig.1 right panel.

2.What’s the validate loss if you scale up the lr for 32M bs in Figure.2? The reason why small batch training can perform better is well studied in previous work,  authors may need to find them out and compare what’s new in the current study.

3.”For example, a batch size of 4M tokens yields the best per-step loss curves, even outperforming the counterparts of larger batch” How about training with more than 3e3 steps? I don’t think bs 4M can outperform bs 32M since bs 32M can eat 8 times more tokens than bs 4M. If not, authors may need to find the better hyper-parameters such as tuning the lr.

**Ethical Concerns:**

["NO or VERY MINOR ethics concerns only"]

**Final Justification:**

I maintain my rating, thanks for the response.

**Limitations:**

yes

**Paper Formatting Concerns:**

No formatting concerns.

**Quality:**

1

**Strengths And Weaknesses:**

Strength:

1.Many experiments are carried out to support the proposed method.

Weakness:

1.Current work just proposes the process commonly used in daily work that scaling the bs after warm-up. Although this work analyzes the gradient noise behavior but it lacks in-depth research on why such phenomena exists and what it means after such behavior? What can we learn from such behavior?

---

> ### Author Rebuttal · Authors · 2025-07-31
>
> Thank you for your review. Regarding your questions:
>
> > Weakness: Current work just proposes the process commonly used in daily work that scaling the bs after warm-up. Although this work analyzes the gradient noise behavior but it lacks in-depth research on why such phenomena exists and what it means after such behavior? What can we learn from such behavior?
>
> We would like to emphasize the contributions of our work:
>
> * First, we want to clarify that the logic of our paper is not to first propose a batch size (BS) scheduler and then support it with other experiments. Instead, we initially observed differences between large and small BS training. Specifically, large BS underperforms small BS in the early stages but can surpass small BS with sufficient training. We believe this phenomenon is not well-known.
>   Given this observation, using a BS scheduler is a straightforward and simple solution to address the early-stage issue of large BS training.
>   Therefore, the BS scheduler itself is not our primary contribution; rather, the key contributions are the observations presented in Figures 1 and 2.
>
> * Second, although many large-scale LLM pretraining technical reports use similar techniques, the use of BS schedulers has not been thoroughly studied in the context of language modeling tasks:
>     - Llama 3 trains on 15T tokens, beginning with a batch size of 4M tokens. They double the batch size to 8M at 252M tokens and then to 16M at 2.87T tokens.
>     - DeepSeek-V3 trains on 14.8T tokens, beginning with a batch size of 12.6M tokens. They gradually increase the batch size from 12.6M to 63M during the first 469B tokens and maintain 63M afterward.
>     - MiniMax-01 does not explicitly specify the total number of tokens in the pretraining stage. Based on their LR schedule in Section 4.2, it appears to be at least 11.4T (=7.2+3.2+1) tokens. They fit a BS scaling law w.r.t. the loss and double the BS whenever the loss reaches this fitted line. Specifically, they double BS from an initial 16M to 32M at 69B tokens, to 64M at 790B tokens, and finally to 128M at 4.7T tokens.
>
>     Except for MiniMax, the BS scheduler used in Llama and DeepSeek is relatively conservative. We speculate they might not fully recognize potential benefits of large BS in pretraining and instead primarily choose large BS for efficiency reasons. However, our experiments demonstrate that increasing BS by 64x via a BS scheduler is viable at the scale of approximately 100B tokens. Additionally, our analysis of optimization-related metrics suggests that the exact length of the BS scheduling phase is not sensitive.
>
>
> > Q1. There’s loss spike shown in the Fig.1 right panel.
>
> We apologize for the misleading figure. Figure 1 was plotted by sampling points directly from the training loss curves without additional smoothing, and training loss is notably noisy, especially at a smaller BS. Therefore, the "loss spike" is merely due to the lack of smoothing in the plot. In fact, there is no actual spike in the loss.
>
> > Q2.1. What’s the validate loss if you scale up the lr for 32M bs in Figure.2?
>
> Using the setting from Figure 2(a), we train for 30B tokens with a BS of 32M, higher LRs including 2^−7.5, 2^−7, and 2^−6.5. Results are presented in Table 1 below, along with the original 32M BS results from Figure 2(a) for clearer comparison. We observe that the optimal LR at a large BS is not necessarily high. Additionally, note that the training runs with LR 2^-7 and 2^-6.5 diverge.
>
> Table 1. Validation loss with BS=32M and various LR in 30B tokens
> | LR |2^-11.5|2^-11|2^-10.5|2^-10|2^-9.5|2^-9|2^-8.5|2^-8| 2^-7.5 | 2^-7 | 2^-6.5 |
> |-|-|-|-|-|-|-|-|-|-|-|-|
> |Loss|**2.524**| 2.630| 2.623| 2.585| 2.620| 2.589 | 2.619| 2.549|2.736|4.978|6.213|
>
> > Q2.2. The reason why small batch training can perform better is well studied in previous work, authors may need to find them out and compare what’s new in the current study.
>
> Many early works [1,2] show that small BS is better and large BS typically hinders model generalization.
> However, there are many differences in experiment setup between those works and our work. Specifically, they mainly focus on multi-epoch CNN image classification with SGD. Our work uses one-pass data in the language modeling task, transformer model, and Adam optimizer. We believe the one-pass or multi-epoch setting is a key factor to the conclusion. Please refer to *the Table 2 in our reply to Reviewer 4vk7* for more details.
>
> On the other hand, many recent works [3,4,5] study the role of BS similar to our setting. Their conclusions are typically like large BS is better as we scale the dataset, which are consistent with our observation.
>
> One exception is the recent work [6], which says small BS is better in languge modeling task. However, they operate in the Chinchilla-optimal setting, which train a model of size N for 20N tokens, while we are in a over-trained setting. As shown in our Figure 1, we believe large BS can outperform small BS with sufficient traing.
>
>
> > Q3. ”For example, a batch size of 4M tokens yields the best per-step loss curves, even outperforming the counterparts of larger batch” How about training with more than 3e3 steps? I don’t think bs 4M can outperform bs 32M since bs 32M can eat 8 times more tokens than bs 4M. If not, authors may need to find the better hyper-parameters such as tuning the lr.
>
> To clarify, Figure 3(c) is intended solely to illustrate the failure of large BS in the **early training stage**: large batch sizes (e.g., 16M, 32M) can fall behind smaller batch sizes (e.g., 2M, 4M), even when compared on a per-step basis. We do not claim that smaller batch sizes like 4M would outperform larger batch sizes like 32M in terms of per-step performance after sufficient training.
>
> As stated at the end of Section 3.1: "Taken together, these results indicate a clear degradation of large-batch optimization in the early stage of training."
>
> [1] On Large-Batch Training for Deep Learning: Generalization Gap and Sharp Minima, ICLR 2017.
>
> [2] AdaBatch: Adaptive Batch Sizes for Training Deep Neural Networks, 2017.
>
> [3] How Does Critical Batch Size Scale in Pre-training? ICLR 2025.
>
> [4] DeepSeek LLM: Scaling Open-Source Language Models with Longtermism, 2024.
>
> [5] Predictable Scale: Part I — Optimal Hyperparameter Scaling Law in Large Language Model Pretraining, 2025.
>
> [6] Small Batch Size Training for Language Models: When Vanilla SGD Works, and Why Gradient Accumulation Is Wasteful, 2025.
>
> ------
>
> Below is the reply to Reviewer haAx due to space limit.
>
> > Q4 by Reviewer haAx. Literature about BS in billion-parameter-scale language model pretraining.
>
> We reviewed how previous works set the BS for large-scale LLM pretraining in **the first part of Section 6 in the paper**. Based on that, we provide a more detailed version here:
>
> - Llama 3 trains on 15T tokens, beginning with a batch size of 4M tokens. They double the batch size to 8M at 252M tokens and then to 16M at 2.87T tokens.
> - DeepSeek-V3 trains on 14.8T tokens, beginning with a batch size of 12.6M tokens. They gradually increase the batch size from 12.6M to 63M during the first 469B tokens and maintain 63M afterward.
> - MiniMax-01 does not explicitly specify the total number of tokens in the pretraining stage. Based on their LR schedule in Section 4.2, it appears to be at least 11.4T (=7.2+3.2+1) tokens. They fit a BS scaling law w.r.t. the loss and double the BS whenever the loss reaches this fitted line. Specifically, they double BS from an initial 16M to 32M at 69B tokens, to 64M at 790B tokens, and finally to 128M at 4.7T tokens.
> - Kimi-K2 trains on 15.5T tokens, maintaining a constant batch size of 67M tokens throughout pretraining.
> - Qwen 2.5/3 mentions using scaling laws to determine batch size but does not provide details.
> Other strong models like GPT-4/4o/4.5, Gemini, Claude, and Grok do not disclose training details.
>
> Some remarks:
>
> - For all models using BS schedulers, the LR schedule is independent of the BS schedule, meaning they do not adjust the LR when increasing BS (e.g., doubling it).
> - **Comparison with our work**: Although some models use larger BS, around 64M or even 128M tokens compared to our experiments, they train on significantly more tokens—**trillions of tokens**. Additionally, except for MiniMax, the BS scheduler used in Llama and DeepSeek is relatively conservative.
> We speculate they might not fully recognize potential benefits of large BS in pretraining and instead primarily choose large BS for efficiency reasons.
> However, our experiments demonstrate that increasing BS by 64x via a BS scheduler is viable at the scale of approximately 100B tokens.

---

> > ### Comment · Reviewer_Q7C1 · 2025-08-06
> >
> > Thanks for the rebuttal.
> >
> > But current work still lacks of novelty, the rebutall will not change my rating.
> >
> > "why such phenomena exists and what it means after such behavior? What can we learn from such behavior?" remains unclear, I hope authors can do some in depth derivations and experiments and resubmit again.
> >
> > This is my final comment.

---

### Official Review · Reviewer_FmKK · 2025-07-02

**Clarity:** 3
**Significance:** 3
**Originality:** 3
**Rating:** 4
**Confidence:** 4

**Summary:**

This paper proposes a batch size scheduler to enable big batch training and show that it can actually lead to better performance than using a fixed batch size (either big or small).

Background: large batch sizes lead to fewer gradient updates and can utilize more resource for distributed training, but batch sizes that are too large hurt performance. This paper establishes that the main reason is that large batch sizes perform poorly at the initial training stage (much worse loss, larger gradient norm).

The authors then designed several different batch size schedulers that linearly increase the batch size. Not only does the scheduler allow bigger "critical batch sizes", but it also leads to better performance (loss) compared to using a fixed batch size, effectively speeding up the training (3.3x reported by the authors).

The authors also conducted additional analysis. They first showed that prior way of estimating critical batch sizes (gradient noise) do not match the empirical finding. They also revealed some curious findings, for example, the Adam update norm (pre-learning-rate) is not associated with learning rates, but batch sizes.

**Questions:**

See the weakness part.

**Ethical Concerns:**

["NO or VERY MINOR ethics concerns only"]

**Final Justification:**

I increased my score as the authors justified their hyperparameter choices with empirical ablations

**Limitations:**

yes

**Quality:**

3

**Strengths And Weaknesses:**

**Strength**

The authors raised a novel finding that large batch sizes mainly fail at the warmup stage and simply increasing the warmup can help stabilize training. The authors then designed a novel batch size scheduler that not only enables larger-batch training (which has system and efficiency gains) but also improves the final loss (performance gains). The analysis also helps us better understand the curious case of language model optimization.

**Weakness**

My two major complains:

(1) In section 3, all experiments have the 2D grid (batch size and learning rate) which is great. However, it seems that the learning rate is fixed when using the batch size scheduler, which seems to be wrong -- shouldn't the learning rate be changed according to the batch size as well? For example, the authors could follow the sqrt scaling law for Adam optimizers following Malladi et al.

Malladi et al. On the SDEs and scaling rules for adaptive gradient algorithms

This is my main complain and I want to hear the authors' clarification/justification on this. I'm willing to raise the score if it is well justified.

(2) It would be great if more evaluation can be done on the main experiment, e.g., adding few-shot downstream task evaluation on common tasks like HellaSwag, MMLU, ARC, etc.

---

> ### Author Rebuttal · Authors · 2025-07-31
>
> Thank you for your constructive review. Regarding your questions:
>
> > Q1. Square root learning rate (LR) - batch size (BS) scaling rule.
>
> This is a good question! We indeed did not explicitly discuss our reasoning behind this choice in the main text.
>
> Frankly, the primary reason is straightforward: in early experiments, we tested the square root scaling rule and found it effective when increasing BS from 0.5M to 8M (with LR increased 4x). However, beyond this point, further increases (like increasing BS to 32M) led to complete training collapse.
>
> Thus, we suspect this rule may not hold for extremely large BS (and therefore very large LR). Below, we provide evidence and discuss our observations:
>
> Previous studies [1,2] proposed a square root scaling rule between LR and BS for Adam. Specifically,
> when increasing BS by Nx, LR should scale by \sqrt{N}x.
> However, understanding this scaling rule and identifying the range in which it holds require further investigation.
>
> We interpret this rule from several perspectives. For language modeling tasks with fixed N and D:
>
> **Perspective 1: Optimal LR at a given BS is proportional to \sqrt(BS).**
> We analyzed smoothed training loss at tokens 5B, 10B, 15B, 20B, 25B, and training end across various LR and BS settings (Table 1 below, based on data from Figure 2 in the paper). We observed:
>
> - During the constant LR stage, the rule initially holds for BS from 0.5M to 2M, extending up to 4M later in training. However, **it fails for BS of 8M or higher.**
> - By the training end (with 0.05x linear LR decay over the last 3B tokens), the scaling rule completely fails, as optimal LR decreases with larger BS. This suggests this rule is sensitive to LR schedule.
>
> However, we'd like to remark that: given observation 1, a key open question is whether the validity range expands with prolonged constant LR training, especially as optimal LR typically decreases over time [3], reducing the risk of LR explosion.
>
>
> Table 1. Optimal LR for various BS and tokens
> | Tokens (B) \ Batch Size | 0.5M    | 1M      | 2M     | 4M    | 8M      | 16M     | 32M     |
> | ------------------- | ------- | ------- | ------ | ----- | ------- | ------- | ------- |
> |5| 2^-10.5 | 2^-10   | 2^-9.5 | 2^-10 | 2^-11.5 | 2^-11.5 | 2^-11.5 |
> |10| 2^-11.5 | 2^-10   | 2^-10  | 2^-10 | 2^-10.5 | 2^-11.5 | 2^-11.5 |
> |15| 2^-11.5 | 2^-10.5 | 2^-10  | 2^-10 | 2^-10.5 | 2^-11   | 2^-11.5 |
> |20| 2^-11.5 | 2^-11   | 2^-10  | 2^-10 | 2^-10.5 | 2^-11   | 2^-11.5 |
> |25| 2^-11.5 | 2^-11   | 2^-10  | 2^-10 | 2^-10.5 | 2^-11   | 2^-11.5 |
> |30 (with LR decay)| 2^-10 | 2^-9.5   | 2^-9.5  | 2^-10 | 2^-10.5 | 2^-11   | 2^-11.5 |
>
>
> **Perspective 2: Square root scaling preserves the loss curve.**
> Actually, this is a stronger condition: if loss curves match exactly under this rule, final optimal hyperparameters naturally follow this rule.
> Experiments in prior work [1] support this viewpoint, even if the hyperparameters where loss curves match are not strictly optimal (since no extensive hyperparameter sweep was conducted, according to [1]). \
> We analyze our training trajectories up to 30B tokens (based on data from Figure 2) and present it in Table 2. We found:
> 1. Loss curves overlap from 0.5M to 4M to some extent, especially during 10B-30B. However, for 8M or higher BS, curves clearly do not match.
> 2. If we are stricter, we observe that during constant LR (before 27B tokens), the 4M curve surpasses the 0.5M curve. We expect that even if training proceeds with constant LR, the gap between the 0.5M and 4M curves will persist, suggesting that the curves might not match eventually.
>
> Table 2. Train loss curve with HPs under the square root scaling rule.
>
> | Tokens (B) \ (BS, LR) | (0.5M, 2^-11.5) | (1M, 2^-11) | (2M, 2^-10.5) | (4M, 2^-10) | (8M, 2^-9.5) | (16M, 2^-9) | (32M, 2^-8.5) |
> | -------------------- | ------ | ------- | ------- | ------- | ------- | -------- | -------- |
> | 4                    | 2.317  | 2.328   | 2.365   | 2.484   | 3.311   | 4.962    | 6.142    |
> | 7                    | 2.239  | 2.241   | 2.253   | 2.304   | 2.645   | 3.705    | 5.245    |
> | 10                   | 2.204  | 2.200   | 2.205   | 2.236   | 2.451   | 3.057    | 4.668    |
> | 13                   | 2.182  | 2.178   | 2.178   | 2.196   | 2.360   | 2.779    | 4.177    |
> | 16                   | 2.168  | 2.158   | 2.157   | 2.168   | 2.304   | 2.621    | 3.630    |
> | 19                   | 2.154  | 2.145   | 2.143   | 2.151   | 2.267   | 2.523    | 3.235    |
> | 22                   | 2.146  | 2.138   | 2.133   | 2.134   | 2.238   | 2.453    | 3.006    |
> | 25                   | 2.139  | 2.128   | 2.123   | 2.126   | 2.218   | 2.403    | 2.856    |
> | 28 (LR is decaying)  | 2.120  | 2.111   | 2.104   | 2.108   | 2.189   | 2.350    | 2.714    |
> | 30                   | 2.064  | 2.062   | 2.061   | 2.068   | 2.148   | 2.300    | 2.630    |
>
>
> **Discussion about related works**:
> - In [1], although they did not specify the range where the scaling rule holds, their experiments showed signs that it starts to fail at BS = 2M (8192*128) (Figure 2(c) in [1]). In the analysis of [2], they explicitly mentioned that square root scaling only holds for relatively small BS. As BS further increases, the optimal LR grows more slowly. Similar findings were reported in other works. For example, Figure 7 in [4] shows that optimal LR stabilizes as BS increases, and Figure 2 (middle) in [5] even shows a decrease in optimal LR. Figure 4 in [6] also shows optimal LR scales more slowly than the square root scaling rule.
> - Additionally, recent technical reports that use BS schedulers (Llama 3, DeepSeek-V3, MiniMax-01) have all chosen to maintain a constant LR. Or more precisely, they use a BS-independent LR scheduler.
>
> In summary, our early experiments show that strictly following the square root scaling rule causes training collapse when increasing BS from 0.5M to 32M. The analysis above suggests that the square root scaling rule gradually does not hold with increasing BS (in our experiments, the threshold was around 8M). However, our main research focus is on large BS (8M and larger), and we remain uncertain about how LR should scale in these cases. Although it’s possible to use square root scaling below 8M and switch to a constant LR beyond that point, we believe 8M is not necessarily a general threshold. Thus, after careful consideration and to simplify our method by minimizing complicating factors, we ultimately decide not to increase LR.
>
> [1] On the SDEs and Scaling Rules for Adaptive Gradient Algorithms, NeurIPS 2022.
>
> [2] Surge Phenomenon in Optimal Learning Rate and Batch Size Scaling, NeurIPS 2024.
>
> [3] Scaling Optimal LR Across Token Horizons, ICLR 2025.
>
> [4] Scaling Law for Language Models Training Considering Batch Size, 2024.
>
> [5] Power Lines: Scaling Laws for Weight Decay and
> Batch Size in LLM Pre-training, 2025.
>
> [6] Small Batch Size Training for Language Models: When Vanilla SGD Works, and Why Gradient Accumulation Is Wasteful, 2025.
>
>
>
> > Q2. More evaluation
>
> Due to time constraints, we could not conduct downstream task evaluations. However, we provide additional results regarding generalization in large BS training in terms of validation loss. Specifically, we add comparisons between training and validation losses for our experiments shown in Figures 1 and 2 and find they closely match. This suggests that large BS does not negatively impact generalization.
> Please refer to *Tables 1 and 2 in our reply to Reviewer haAx for more details.*

---

> > ### Comment · Reviewer_FmKK · 2025-08-08
> > **Thanks for the additional results**
> >
> > Thanks for the additional results! I also saw relevant discussions in the other thread about beta, which is also essential. I guess in this case, the only rigorous way is to actually sweep these hyperparameters (maybe using the sqrt scaled lr/beta as a baseline), which I can imagine to be super expensive. With the additional discussion and results, I'm wiling to increase my score. Please include these additional discussions in the final version.

---

### Official Review · Reviewer_haAx · 2025-07-03

**Clarity:** 3
**Significance:** 2
**Originality:** 2
**Rating:** 3
**Confidence:** 4

**Summary:**

- This paper presents the observation that, when pretraining a 164M language model, large-batch training can achieve lower training loss than small-batch baselines when the training happens on sufficiently many tokens. But large-batch training exhibits unstable gradient norms and worse per-step loss during warm-up.
- To resolve this challenge, the paper proposes a simple batch size scheduler so that the batch size is small at the beginning and large later. In one experiment, using the scheduler to anneal from batch size 0.5M to 8M achieves the similarly low training loss with 3.3 times less tokens than always using 8M batch size.
- The paper observes that different batch size schedules converge to similar final losses. It also presents the finding that the optimal batch size cannot be predicted by classic gradient noise scale metrics.

**Questions:**

- The paper reports training loss in Figure 1. Could you report that for validation loss also? Reaching a low loss on a held-out set may be more important than overfitting the training set.
- The paper reports that large-batch training exhibits lower training loss given sufficient amounts of tokens process. Do these have to all be different data? Or does the advantage also happen given multiple epochs of the same data?
- L59-L66 discusses literature about large-batch training hindering model generalization, which seems to remain as an issue. Are you saying that given sufficient training, large-batch training actually does not hinder model generalization? Could you explain and provide pointers to evidence?
- The paper experiments with 164M language models. This is understandable due to computational constraints. However, it would be great if the authors can discuss the literature on billion-parameter-scale language model pretraining. What batch size have state-of-the-art language models used?
- You found that a large batch size does not work well at the warmup stage. Is it because of the small learning rate, the randomly initialized parameters, or both? Assume there’s no warmup; how does large vs. small batch size compare in the early training stage? Assume the warmup stage is short, should the small batch size still be used for some more steps after warmup?
- In Figure 3, you observe that 4M batch size is better than 0.5M and 32M. It’s understandable that too small and too big of a batch size will both not work well. Have you tried to plot Figure 3 for the later training stage to see what’s an optimal batch size for the later stage? Or did you find that increasing the batch size keeps improving training and validation loss in the later stage?

**Ethical Concerns:**

["NO or VERY MINOR ethics concerns only"]

**Final Justification:**

I have carefully reviewed the rebuttal. Since multiple concerns remains, I will keep my score.

**Limitations:**

Limitations are discussed. Potential negative societal impacts seem to be missing, although it's not apparent that this work has any immediate ones.

**Quality:**

3

**Strengths And Weaknesses:**

- The paper is well motivated. Small batch sizes are needed early in the training and big batch sizes are advantageous later, and thus the paper proposes a batch size scheduler. The solution is straightforward and experiments are rigorous. The paper is well written also.
- There are multiple questions to be addressed in order that the investigation is more convincing and impactful. A main issue is that the paper often reports training loss instead of generalization metrics. Please refer to the Questions section below.

---

> ### Author Rebuttal · Authors · 2025-07-31
>
> Thank you for your careful review. Regarding your questions:
>
> > Q1. Validation loss.
>
> To further support our observations, we report validation loss for the experiments shown in Figures 1 and 2 in Tables 1 and 2, respectively:
> - In Table 1, we report validation loss at various checkpoints for each trajectory in Figure 1. We observe that the trends, relationships between curves, and validation loss values closely match training loss in Figure 1.
> - In Table 2, we present the data from Figure 2(a) as heatmaps in Tables 2(a). Additionally, we report the corresponding training losses for these runs in Tables 2(b). Across all runs in the grid, we observe:
>     - Training and validation loss values are similar.
>     - The optimal LR for each BS is consistent between training and validation losses.
>
> In fact, throughout all experiments presented in our paper, validation losses remain at similar levels to the training losses. For loss curves, we present training loss, as it may more precisely reflect training dynamics. For grid search results, especially in Figure 2, we present validation loss since the noise is smaller.
>
> Table 1. Validation loss associated to Figure 1.
>
> | Tokens (B) \ BS |8M|0.5M|BS scheduler (0.5M -> 8M)|
> |-|-|-|-|
> |8.52|–|–| 2.219|
> |8.39|2.392|2.217|–|
> |14.42|–|–| 2.186|
> | **15.07** *(BS growth begins)*|–|–| **2.182**|
> |16.78|2.228|2.172|–|
> |17.30|–|–| **2.139** *(sharp drop begins)* |
> |22.39|–|–| **2.076**|
> |25.17|2.166|2.151| –|
> | **28.61** *(BS growth is ending)*| –| – | **2.049**|
> | **37** *(large BS surpass small BS)* | **2.124** | 2.144| – |
> |38.70| – | – | 2.036 |
> |45| 2.110| 2.141| – |
> |52| 2.097| 2.133| – |
> |59.67| – |– |2.027|
> |68| 2.079|2.130| -|
> |80.64| –| – | 2.024|
> |84| 2.070| 2.125|–|
> | 91.13| –| – | 2.024|
> | 92 | 2.062 | 2.122|-|
> | 100| 2.061 | 2.119| -|
> | 101.61 | –| – | **2.018**|
>
> Table 2. A heatmap version of Figure 2 (train loss and validation loss).
>
> (a) final validation loss using constant BS
> | BS \ LR *(Val loss, const BS)*|2^-11.5|2^-11|2^-10.5|2^-10|2^-9.5|2^-9| 2^-8.5|2^-8|
> | --------------- | --------- | ------- | --------- | ------- | -------- | ------ | -------- | ------ |
> |0.5M| 2.063| 2.061| **2.057** | **2.057**| **2.057**| 2.061| 2.066 | 2.077 |
> |1M| 2.073| 2.064| 2.059| 2.055| **2.053**| 2.056| 2.058 | 2.062|
> |2M| 2.084| 2.073| 2.065| 2.058|**2.055**| 2.056| 2.063 | 2.083|
> |4M| 2.116| 2.096| 2.085|**2.075**| 2.085 | 2.093| 2.105 | 2.118|
> |8M| 2.179| 2.151|**2.133**| 2.146| 2.152 | 2.157| 2.157 | 2.206|
> |16M| 2.290|**2.250**| 2.288| 2.263| 2.271 | 2.298| 2.317 | 2.302|
> |32M|**2.539**| 2.645| 2.638| 2.600| 2.635 | 2.604| 2.634 | 2.564|
>
> (b) final train loss using constant BS
> | BS \ LR *(Train loss, const BS)*|2^-11.5|2^-11|2^-10.5|2^-10|2^-9.5|2^-9|2^-8.5|2^-8|
> |-|-|-|-|-|-|-|-|-|
> |0.5M|2.061|2.060|2.057|2.055|**2.054**|2.057|2.063|2.073|
> |1M|2.071|2.063|2.058|2.052|**2.051**|2.055|2.056|2.059|
> |2M|2.088|2.077|2.068|2.062|**2.058**|2.060|2.067|2.086|
> |4M|2.111| 2.091|**2.070**| **2.070** |2.080|2.088|2.094|2.107|
> |8M|2.185|2.157|**2.138**| 2.151| 2.158| 2.162 | 2.162|2.212|
> |16M|2.286|**2.246**| 2.285| 2.260|2.268| 2.295 | 2.314|2.299 |
> |32M|**2.524**| 2.630| 2.623| 2.585| 2.620| 2.589 | 2.619| 2.549 |
>
> > Q2. One-pass vs. multi-epoch data.
>
> To compare the one-pass and multi-epoch setting, we conduct experiments on a 3B subset of pile and train in multiple epochs. We use the same settings as in Figure 1 (fixed LR of 2^-10, BS 0.5M and 8M). Due to space constraint, please refer to the *Table 2 in the reply to Reviewer 4vk7* for details.
> We can see:
>
> - Large BS initially underperforms but eventually surpasses small BS in both one-pass and multi-epoch settings.
> - The performance gap shrinks under multi-epoch conditions, which suggests dataset reuse might influence benefits of large BS.
>
> Based on these experiments, we do observe the multi-epoch training can make some differences: the benefit of large BS become less pronounced inmulti-epoch setting. However, since the optimal HPs in the multi-epoch setting may differ from the one-pass setting, we cannot currently make a definitive claim regarding the impact of BS.
>
> > Q3. Does large-batch training not hurt generalization given sufficient training, contrary to prior literature?
>
> - First, we want to clarify that in the discussion below, we only use validation loss as the metric for measuring generalization. Some prior works indicate validation loss may not fully represent a model’s capability across other benchmarks.
> - In L59-L66, we discuss how prior works typically view large batch training. They often state that large batch training can hinder model generalization. More precisely, we interpret this as *large BS achieving higher validation loss compared to small BS at the same training loss.*
> - In our experiments, we find that large BS can surpass small BS with sufficient training in terms of both training and validation loss, and these two losses within a single run remain very close (as shown in Q1). Thus, our conclusion is: **Yes, large batch training does not actually hinder model generalization.**
> - The reason why our observations contradict prior works lies in differences in experimental setups. Specifically, differences include model, data, and optimizer. Although we cannot fully isolate all these factors, we believe that training with limited data is a key factor. Evidence supporting this is provided in Q2.
>
> > Q4. Literature about BS in billion-parameter-scale language model pretraining.
>
> **Due to space constraint, we answer this question in the reply to Reviewer Q7C1.**
>
> > Q5. You found that a large batch size does not work well at the warmup stage. Is it because of the small learning rate, the randomly initialized parameters, or both? Assume there’s no warmup; how does large vs. small batch size compare in the early training stage? Assume the warmup stage is short, should the small batch size still be used for some more steps after warmup?
>
> We believe that the poor performance of large BS during warmup primarily arises from **randomly initialized parameters**. This insight can be supported by observations in Figure 2 of our paper:
>
> - Large BS performs significantly better when starting from a post-warmup checkpoint rather than from random initialization.
> - Moreover, improvements for large BS are consistent across all LRs tested, indicating that the benefit of initializing from a post-warmup checkpoint persists even at smaller LR. This strongly suggests random initialization is the key factor.
>
> Regarding the last question, due to time constraints, we haven't explored joint tuning of warmup length and the length of using small BS beyond warmup. However, we can provide some insights based on our experiments:
> - Firstly, we recommend avoiding excessively short warmup stages, as indicated by **Table 3 in our reply to Reviewer tybB**, where warmup length clearly impacts final performance.
> - Secondly, Figures 5 and 6 from the paper show that once training switches to a large batch size, optimization-related statistics eventually converge. Therefore, the use of small BS is to ensure the good performance in the early stage, while how many steps we use in small BS become less pronounced over a sufficiently long training period.
>
>
> > Q6. Have you tried to plot Figure 3 for the later training stage to see what’s an optimal batch size for the later stage? Or did you find that increasing the batch size keeps improving training and validation loss in the later stage?
>
> In Table 3, we present the validation loss for various BS across approximately 100B tokens of training. Using this table, we can identify the optimal BS for a given training budget. Due to computational constraints, we used a fixed LR of 2^-10. We clearly observe the following:
>
> - **As training proceeds, the optimal BS increases.** Initially, it is 0.5M at 2.6B tokens (around the Chinchilla-optimal point), and eventually reaches 4M at 100B tokens.
> - **Large BS outperforms smaller BS in the middle and later training stages.** Specifically, BS=8M surpasses 0.5M at 29B tokens, surpasses 1M at 45B tokens, and surpasses 2M at 92B tokens. Additionally, BS=16M surpasses 0.5M at 60B tokens and surpasses 1M at 100B tokens. Thus, within our observed data range, we confirm that increasing the batch size continuously improves training and validation loss in later stages.
> - The optimal BS reaches 4M at 100B tokens, but we note the gap between 4M and 8M is narrowing (from 0.02 at 45B tokens to 0.006 at 107B tokens). We suspect the optimal BS might shift to 8M if training continues further. However, considering (1) our selected BS intervals are log-uniform, and (2) there is an irreducible loss, it is reasonable to expect the shift from 4M to 8M to take longer compared to the shift from 2M to 4M.
>
> In summary, our results support the conclusion that larger batch sizes improve training and validation loss in the later stages.
>
>
> Table 3. Validation loss for various BS with constant LR.
>
> |Tokens (B) \ BS |0.5M|1M|2M|4M|8M|16M|32M|
> | -------------------- | ------- | ----- | ----- | ----- | ----- | ------ | ------ |
> | 2.6| **2.350** | 2.378 | 2.464 | 2.651 | 3.170 | 4.615 | 5.953 |
> | 10.5  | 2.208 | **2.188** | 2.196 | 2.230 | 2.321 | 2.606 | 4.067 |
> | 29 | 2.155| 2.127 | **2.113** | 2.123 | 2.151 | 2.238| 2.624|
> | 37 | 2.144| 2.116 | **2.101** | 2.102 | 2.124 | 2.194| 2.475|
> | 45 | 2.141| 2.112 | 2.097 | **2.091** | 2.110 | 2.166| 2.380|
> | 52 | 2.133| 2.104 | 2.090 | **2.084** | 2.097 | 2.142| 2.319|
> | 60 | 2.135| 2.100 | 2.086 | **2.073** | 2.096 | 2.123| 2.304|
> | 68 | 2.130| 2.096 | 2.073 | **2.070** | 2.079 | 2.114| 2.258|
> | 76 | 2.126| 2.087 | 2.074 | **2.064** | 2.074 | 2.100| 2.227|
> | 84 | 2.125| 2.090 | 2.067 | **2.061** | 2.070 | 2.094| 2.208|
> | 92 | 2.122| 2.082 | 2.065 | **2.054** | 2.062 | 2.084| 2.198|
> | 100| 2.119| 2.082 | 2.068 | **2.047** | 2.061 | 2.081| 2.178|
> | 107| 2.118| 2.081 | 2.059 | **2.049** | 2.055 | 2.070| 2.166|

---

> > ### Comment · Reviewer_haAx · 2025-08-06
> >
> > - I appreciate the authors’ efforts in putting together the detailed rebuttal, but most of my concerns remain unsolved. In addition, after reading other reviewer’s comments, I was reminded that changing batch size during pretraining is a common practice and thus the proposal of a linear batch size scheduler is not technically very novel.
> > - Q1: It is good to see that results on the validation loss show the same pattern as the training loss and the claims.
> > - Q2: The authors indicate that no definitive claims regarding the impact of batch size can be made according to the authors. This is concerning since the paper’s main claims are about the impact of batch size on the model performance.
> > - Q3: The authors are not able to conclude why the finding in this paper is contradictory to those in the literature.
> > - Q4: The contributions are limited by the fact that experiments are conducted on 164M parameter models. It’d unclear how the findings will apply to billion-scale models.
> > - Q5: The paper interweaves the discussion of warmup and batch size. For example, Section 3.2 looks at the batch size during the warmup phase. However, the relation between both remains unclear. Since the authors suggest that the poor performance of large batch size at the beginning of the training arises from randomly initialized parameters, it’d be good to isolate the impact of warmup.
> > - Q6: This question is addressed.

---

> ### Author Response · Authors · 2025-08-06
>
> > Q3: The authors are not able to conclude why the finding in this paper is contradictory to those in the literature.
>
> Yes, we acknowledge that we cannot precisely conclude why the findings in this paper differ from those in the existing literature. However, we would like to respectfully highlight the following points:
> - We primarily focus on the impact of batch size in the language modeling task using a modern pipeline. Our observation -- that large batch size does not negatively impact generalization -- is itself a valuable contribution.
> - The experimental settings in previous studies differ significantly from ours and are thus not our main focus. We agree that an important next step is to systematically ablate these differences to identify key factors contributing to the contrasting conclusions.
> - From another perspective, our work demonstrates that the findings in language modeling differ considerably from those previously reported. We believe this is valuable for advancing the community's understanding of large batch training.
>
> > Q4: The contributions are limited by the fact that experiments are conducted on 164M parameter models. It’d unclear how the findings will apply to billion-scale models.
>
> - Firstly, we would like to mention that we have results of a 1B model, presented in the rebuttal to Reviewer 4vk7 (Table 1 therein). We emphasize that **our main conclusions remain valid for the 1B model.** Although 1B parameters is not large in the current era, it is not small for optimization or scaling law studies. Many published works use 1B [1,2,3] as their largest model size.
> - Secondly, we highlight that we study an over-trained setting, whereas many other works study the Chinchilla-optimal setting.
>     - On one hand, things can be very different if we train more tokens. As shown in Figure 1, if we train only to 3B tokens (approximately the Chinchilla-optimal setting), large batch size lags behind small batch size. However, large batch sizes demonstrate an advantage starting from 30B tokens, precisely in an over-trained setting.
>     - On the other hand, even if we focus on a small model of 164M in our main text, we use a large amount of tokens, thus considerable compute, to train it. Specifically, Figure 1 involves training with 100B tokens, equivalent to approximately 1.6e19 FLOPS of compute. This amount of compute is significant and approaches the compute budget that others use to train a 1B model.
>
> > Q5: The paper interweaves the discussion of warmup and batch size. For example, Section 3.2 looks at the batch size during the warmup phase. However, the relation between both remains unclear. Since the authors suggest that the poor performance of large batch size at the beginning of the training arises from randomly initialized parameters, it’d be good to isolate the impact of warmup.
>
> Thank you for your suggestion! We first would like to clarify that we attribute the poor performance of large batch sizes at the beginning of training primarily to random initialization. Since warmup typically occurs at the very start of training, we actually refer warmup to the early training stage.
>
> To be more precise, warmup is a phase designed to gradually increase the learning rate and can, in principle, be decoupled from random initialization.
> However, it is important to note that isolating the impact of warmup from initialization is not straightforward, since stability issues may arise without using warmup.
>
> In summary, we agree that the effects of random initialization and warmup can be further decoupled, and we will carefully explore each in future work.
>
>
> [1] Resolving Discrepancies in Compute-Optimal Scaling of Language Models, NeurIPS 2024.
>
> [2] How Does Critical Batch Size Scale in Pre-training? ICLR 2025.
>
> [3] Small Batch Size Training for Language Models: When Vanilla SGD Works, and Why Gradient Accumulation Is Wasteful, 2025.

---

### Comment · Area_Chair_vA5h · 2025-08-04

Dear reviewers, this paper needs a discussion. Could you please reply to the authors asap?

---

### Note · Authors · 2025-08-13

Dear AC and Reviewers,

Thank you for the thoughtful feedback. We appreciate the reviewers’ many positive notes (e.g., novelty, rigorous experiments, clear writing) and constructive review. Below are our final remarks.

**Core finding.**
- Large batch sizes (BS) underperform small BS in the early stage but can surpass them with sufficient training (up to 100B tokens).
- The failure of large BS in early stage is consistent across LR–BS sweeps, and supported by per-step loss and gradient norm, and is not fixed by longer warmup; however, using a small BS only during warmup markedly improves early optimization and overall compute efficiency.

**Practical takeaway.** A simple BS scheduler using small BS early and large BS later improves training:
- At BS=8M, we match the performance of BS=0.5M using ~3.3x less compute.
- At BS=32M, naive training fails at 100B tokens, while the scheduler enables training.

**Analysis.** Dynamics of optimization metrics explain why different BS schedules align in the late stage.

**New experiments in the rebuttal.**
- LR schedule: With WSD decay, the core finding still holds. (T1 to tybB.)
- beta2 tuning: Increasing beta2 to favor small BS does not overturn the core finding. (T2 to tybB)
- Larger scale: Results replicate at larger scale (e.g., 1B models at 30B tokens; 164M at 100B across multiple BS). (T1 to 4vk7, T3 to haAx)

**Clarifications.**
- Square-root LR scaling degrades as BS grows (both optimal LR and loss-curve matching) (T1,T2 to FmKK); at BS=32M, larger LR choices diverge (T1 to Q7C1).
- Generalization: Validation loss curves across settings are close to train loss, show large BS does not harm generalization in our one-pass LM setup. (T1, T2 to haAx)
- Compute & regime: Although 164M is modest, our experiments use up to ~1e19 FLOPs and operate in the over-trained regime, not only the Chinchilla-optimal regime common in prior work.

**Relation to prior findings.**
- Contrary to prior belief, in one-pass language modeling we observe no harm to generalization from large BS (in terms of validation loss).
- Prior work often use substantially different setups. Because our focus is one-pass LM, we do not attempt to reconcile all regimes and cannot precisely attribute the discrepancy.
- Preliminary multi-epoch runs suggest the advantage of large BS is less pronounced there, reinforcing that conclusions are regime-dependent. (T2 to 4vk7)

We thank the AC and reviewers for their time and consideration.

Sincerely,

Authors

---

### Decision · Program_Chairs · 2025-09-17

**Decision:**

Reject

**Comment:**

This paper challenges the existing wisdom that large batch sizes hinder language model training by arguing that the observed underperformance is actually caused by poor behavior during the warmup phase, and consequently proposes a novel warmup strategy that achieves superior results.

The reviewers and the authors engaged in a discussion, but the scores remained low, at the level that does not warrant acceptance for this conference. The reviewers recommended some additional work that would make this work stronger, and the authors would be encouraged to implement the recommended changes. A journal may also be a better option for this work.